

# A Nitrate Ion Chemical Ionization Atmospheric Pressure interface Time-of-Flight Mass Spectrometer (NO₃⁻ ToFCIMS): calibration and sensitivity study

Stéphanie Alage[1], Vincent Michoud[2], Sergio Harb[1], Bénédicte Picquet-Varrault[1], Manuela Cirtog[1], Avinash Kumar[3], Matti Rissanen[3,4], and Christopher Cantrell[1]

[1] Univ Paris Est Creteil and Université de Paris Cité, CNRS, LISA, F-94010 Créteil, France

[2] Université Paris cité and Univ Paris Est Creteil, CNRS, LISA, F-75013 Paris, France

[3] Aerosol Physics Laboratory, Physics Unit, Faculty of Engineering and Natural Sciences, Tampere University, 33720 Tampere, Finland

[4] Chemistry Department, University of Helsinki, 00014 Helsinki, Finland

*Correspondence to*: (Vincent Michoud: Vincent.michoud@lisa.ipsl.fr)

**Abstract.** Volatile organic compounds (VOCs) play a key role in tropospheric chemistry, giving rise to secondary products such as highly oxygenated organic molecules (HOMs) and secondary organic aerosols (SOA). HOMs, a group of low-volatility gas-phase products, are formed through the autoxidation process of peroxy radicals ($RO_2$) originating from the oxidation of VOCs. The measurement of HOMs is made by a $NO_3^-$ ToFCIMS instrument, which also detects other species like small highly oxygenated VOCs (e.g. dicarboxylic acids) and sulfuric acid ($H_2SO_4$). The instrument response to HOMs is typically estimated using $H_2SO_4$, as HOMs are neither commercially available nor easily synthesized in the laboratory. The resulting *calibration factor* is then applied to quantify all species detected using this technique. In this study, we explore the sensitivity of the instrument to commercially available small organic compounds, primarily dicarboxylic acids, given the limitations associated with producing known amounts of HOMs for calibration. We compare these single compound calibration factors to the one obtained for $H_2SO_4$ under identical operational conditions. The study found that the sensitivity of the $NO_3^-$ ToFCIMS varies depending on the specific type of organic compound, illustrating how a single calibration factor derived from sulfuric acid is clearly inadequate for quantifying all detected species using this technique. The results highlighted substantial variability in the calibration factors for the tested organic compounds, with 4-nitrocatechol exhibiting the highest sensitivity, and pyruvic acid the lowest. The obtained sulfuric acid calibration factor agreed well with the previous values from the literature. In summary, this research emphasized the need to develop reliable and precise calibration methods for progressively oxygenated reaction products measure with $NO_3^-$ CIMS, for example, HOMs.

## 1 Introduction

Volatile organic compounds (VOCs), originating from both natural sources (such as land or marine ecosystems, also known as biogenic VOCs or BVOCs) and human-made sources (referred to as anthropogenic VOCs or AVOCs), play crucial





roles in tropospheric chemistry. Once released into the atmosphere, VOCs undergo chemical oxidation reactions initiated primarily by the three main atmospheric oxidants: hydroxyl radicals (OH·), nitrate radicals (NO$_3$·), and ozone (O$_3$) (Finlayson-Pitts, 2010). Such chemical reactions ultimately result in the production of carbon dioxide (CO$_2$) but also may lead to the formation of multifunctional organic compounds, although this is a minor pathway. These are typically less volatile than the initial compounds, except in cases of fragmentation, and can therefore take part in the formation and growth of secondary organic aerosols (SOA) (Kanakidou et al., 2005; Riipinen et al., 2011). Consequently, VOCs are well-recognized as important precursors for the formation and growth of SOA as well as other secondary products, such as ground-level ozone, which significantly impact air quality, human health, and climate (Seinfeld & Pandis, 2016).

Recently, studies have revealed that AVOCs and BVOCs play crucial roles as key precursors in the gas-phase formation of Highly Oxygenated organic Molecules (HOMs). Initially labeled as Extremely Low Volatile Organic Compounds (ELVOCs) to emphasize their pivotal role in particle formation and growth (Schobesberger et al., 2013; Ehn et al., 2012, 2014; Jokinen et al., 2015), it has since been suggested that these compounds span a broader range of volatility classes (Kurtén et al., 2016), including ULVOCs (Ultra Low Volatile Organic Compounds), ELVOCs, LVOCs (Low Volatile Organic Compounds), and SVOCs (Semi Volatile Organic Compounds). This recognition acknowledges their ability to contribute to gas-to-particle partitioning with varying levels of efficiency (Donahue et al., 2011; Kirkby et al., 2016; Tröstl et al., 2016; Bianchi et al., 2019; Guo, Yan, et al., 2022).

The term "HOM**s**" primarily refers to a class of products generated through the gas-phase chemical process known as autoxidation (Bianchi et al., 2019; Berndt et al., 2021). The significance of autoxidation in atmospheric chemistry only gained recognition in the past decade (Crounse et al., 2013; Ehn et al., 2014; Rissanen et al., 2014). Overall, it is characterized by an intramolecular H-atom shift within peroxy radicals (RO$_2$·), which are formed following the initial reaction of VOCs with oxidants, yielding a hydroperoxide functionality (HOO-). This is followed by rapid addition of O$_2$ to form a new, more oxygenated RO$_2$·. This process can repeat multiple times (Crounse et al., 2013; Ehn et al., 2017; Møller et al., 2019; Vereecken & Nozière, 2020). Autoxidation may also be interrupted at each stage by classical termination reactions (unimolecular or bimolecular reactions). This evolution depends largely on the chemical environment. These reactions convert HOM-RO$_2$· into closed-shell molecules, while preserving the number of carbon atoms. In some instances, this interruption can lead to the formation of RO· radicals, which subsequently contribute to the production of closed-shell molecules. Due to their low volatility, HOMs are expected to efficiently partition into the particle phase. Consequently, they have the potential to condense onto existing aerosols or contribute to new particle formation (NPF) (Ehn et al., 2014; Riccobono et al., 2014; Kirkby et al., 2016).

Following their initial observation in the atmosphere as ambient ion clusters with the nitrate ion (NO$_3^-$), nitrate chemical ionization time-of-flight mass spectrometry (nitrate ToFCIMS) has been employed to detect neutral HOMs in the atmosphere and in laboratory settings (Ehn et al., 2012; Jokinen et al., 2014). Other reagent ions for detecting HOMs, such as acetate and iodide, have also been used (e.g. Berndt et al., 2016; Iyer et al., 2017; Hansel et al., 2018; Riva et al., 2019).





The nitrate-ion based chemical ionization atmospheric pressure interface time-of-flight mass spectrometer ($NO_3^-$
ToFCIMS; Aerodyne Research Inc., and Tofwerk AG) is the key online mass spectrometric (MS) instrument characterised
towards organic compound detection in this study. This instrument is capable of measuring gas-phase non-radical HOMs,
highly oxidized peroxy radicals (HOM-$RO_2\cdot$), certain oxygenated VOCs (OVOCs, such as small dicarboxylic acids), and
sulfuric acid ($H_2SO_4$) with high resolution and sensitivity (Bianchi et al., 2019; Ehn et al., 2017; Rissanen, 2021). A concise
description of the instrument's principles and the operational parameters used will be provided in the following section.

For the quantification of HOMs, as well as other organic compounds detected by the ToFCIMS, under typical
sampling conditions, the following general Eq. (1) is employed:

$$[\mathbf{X}] = \mathbf{C_X} \times \frac{\mathbf{i X^-} + \sum_{n=0-2} \mathbf{i_{HX(HNO_3)_n NO_3^-}}}{\mathbf{i_{NO_3^-}} + \mathbf{i_{(HNO_3)NO_3^-}} + \mathbf{i_{(HNO_3)_2 NO_3^-}} + \mathbf{i_{(H_2O)NO_3^-}}} \,, \tag{1}$$

Where, $\mathbf{C_x}$ is the calibration coefficient (molecules $cm^{-3}$ $ncps^{-1}$; ncps : normalized counts per second), $[\mathbf{X}]$ is the concentration
of the measured compound by the ToFCIMS, $\mathbf{i_{X^-}}$ is the ion signal of the deprotonated product ion (typically for acidic organic
compounds), $\sum_{n=0-2} \mathbf{i_{HX(HNO_3)_n NO_3^-}}$ is the sum of the product ion cluster signals, and $\mathbf{i_{NO_3^-}} + \mathbf{i_{(HNO_3)NO_3^-}} + \mathbf{i_{(HNO_3)_2 NO_3^-}} +$
$\mathbf{i_{(H_2O)NO_3^-}}$ is the sum of reagent ion signals for $NO_3^-$, $(HNO_3)\cdot NO_3^-$, $(HNO_3)_2\cdot NO_3^-$ and $H_2O\cdot NO_3^-$.

The presence of $H_2O\cdot NO_3^-$ clusters was identified in the mass spectra, and their response showed variations with changing
humidity conditions during sampling. Consequently, their signals were incorporated into the calculations using Eq. (1),
although their influence was found to be relatively minor, typically accounting for 0.1-2% of the total reagent ion signal.

Calibrating the $NO_3^-$ ToFCIMS is essential to determine appropriate calibration coefficient, $C_x$, which reflects
specific instrument's detection sensitivity to an organic compound, denoted as X, and can be sued to determine the molecule's
concentration [X]. It is important to note that it's almost certainly impossible to derive a single calibration factor capable of
evaluating all possible organic compounds. The $NO_3^-$ ToFCIMS can detect a broad range of low volatility multifunctional
organic species, and finding suitable oxygenated organic standards for calibration is challenging, especially considering the
complexity and limited knowledge of HOMs' precise structures. To determine the appropriate $C_x$ values, a series of known
concentration of a compound must be sampled. The $C_x$ value is then obtained as the slope of the plot illustrating the known
concentration as a function of the normalized ion product signals.

The most commonly employed method in the literature relies on using sulfuric acid as a calibrant, assuming that it
exhibits the same ionization kinetic rate constant and comparable transmission efficiency as HOMs (Ehn et al., 2014; Kirkby
et al., 2016). Employing a $C_X$ value determined for $H_2SO_4$ introduces considerable uncertainty into the obtained values. In
brief, calibrations using sulfuric acid are typically conducted by generating a known quantity of OH· in an excess of $SO_2$
leading to a known amount of gas-phase sulfuric acid (Berndt et al., 2014). Most studies are based on a dedicated set-up
developed to calibrate a $NO_3^-$ CIMS with $H_2SO_4$. This system was estimated to give an overall uncertainty of around 33%
(Eisele & Tanner, 1993; Kürten et al., 2012). Furthermore, Jokinen et al. (2012) presented $C_X$ values obtained by comparing





$H_2SO_4$ measured in ambient air by a ToFCIMS to concentrations measured by a calibrated quadrupole CIMS. Other direct calibrations have been reported in the literature, using alternative organic compounds such as perfluorinated heptanoic acid $C_7HF_{13}O_2$ (Ehn et al., 2014), malonic acid $C_3H_4O_4$ (Krechmer et al., 2015; Isaacman-VanWertz et al., 2018; Massoli et al., 2018), and 4-nitrophenol $C_6H_5NO_3$ (Cheng et al., 2021). It is noteworthy that many studies use previously determined $C_X$ values from the literature introducing potentially larger uncertainties into their measurements (Zha et al., 2018; Wang et al., 100  2020; Garmash et al., 2020; Meder et al., 2022; Zhang et al., 2022). Despite being obtained under varying experimental conditions, the reported calibration factors generally exhibit a similar order of magnitude in the range of $(0.2-6) \times 10^{10}$ molecule $cm^{-3}$ $ncps^{-1}$ (Table 1).

**Table 1: Calibration factors from the literature using $NO_3^-$ ToFCIMS instruments.**

| Reference | Calibration coefficient (molecule $cm^{-3}$ $ncps^{-1}$) | Calibrant |
|---|---|---|
| **Jokinen et al., 2012** | $5 \times 10^9$; $1.89 \times 10^{10}$ | $H_2SO_4$ |
| **Rissanen et al., 2014** | $1.94 \times 10^{10}$ | $H_2SO_4$ |
| **Berndt et al., 2015, 2016; Jokinen et al., 2014, 2015** | $1.85 \times 10^9$ | $H_2SO_4$ |
| **Mutzel et al., 2015** | $8.4 \times 10^9$ | $H_2SO_4$ |
| **Kirkby et al., 2016** | $6.5 \times 10^9$ | $H_2SO_4$ |
| **Kürten et al., 2016** | $6 \times 10^9$ | $H_2SO_4$ |
| **Riva, 2019** | $2 \times 10^{10}$ | $H_2SO_4$ |
| **Quéléver et al., 2019** | $1.65 \times 10^9$ | $H_2SO_4$ |
| **Pullinen et al., 2020** | $3.7 \times 10^{10}$ | $H_2SO_4$ |
| **Shen et al., 2021 Zhao et al., 2021 Guo et al., 2022 Luo et al., 2023** | $2.5 \times 10^{10}$ | $H_2SO_4$ |
| **Cheng et al., 2021** | $1.66 \times 10^{10}$ | $H_2SO_4$ |
| **Xu et al., 2021** | $1.57 \times 10^{10}$; $2 \times 10^{10}$ | $H_2SO_4$ |
| **Dam et al., 2022** | $6 \times 10^{10}$ | $H_2SO_4$ |
| **Wang et al., 2022** | $1.1 \times 10^{10}$ | $H_2SO_4$ |
| **Ehn et al., 2014** | $1.6 \times 10^{10}$ | $C_7HF_{13}O_2$ |
| **Krechmer et al., 2015 Massoli et al., 2018** | $7.9 \times 10^{10}$ | $C_3H_4O_4$ |
| **Cheng et al., 2021** | $1.62 \times 10^{10}$ | 4-nitrophenol |

In this paper, we tested several commercially available organic compounds as potential direct calibrants for the $NO_3^-$ 105  ToFCIMS instrument. We describe and discuss the implementation of two calibration approaches in sub-sections 2.2.1 and 2.2.2. Additionally, we established a method to experimentally determine the vapor pressure (Pvap) of specific solid organic compounds, including malonic acid, detailed in sub-section 2.2.2. Finally, we discuss instrument calibration using sulfuric acid and compare the results to those obtained with organic compounds.

## 2 Materials and Experimental Methods

### 2.1 The nitrate ToFCIMS

The $NO_3^-$ ToFCIMS used in this study is composed of two primary components: the chemical ionization (CI) inlet (Eisele & Tanner, 1993; Jokinen et al., 2012) and the atmospheric pressure interface time-of-flight mass spectrometer (APi-





ToF) (Junninen et al., 2010). Briefly, nitrate reagent ions $(HNO_3)_{n=0\text{-}2}\cdot NO_3^-$ are generated by passing a 30 sccm (standard cm$^3$.min$^{-1}$) stream of dry air through a small amount (2-20 mL) of liquid nitric acid (HNO$_3$) placed in a glass vial producing a

mixture of gas-phase HNO$_3$ in air (about 6% by volume). This flow is mixed with the sheath flow and then exposed to soft X-ray radiation (Hamamatsu Photoionizer Model C12646 power supply; Model L9491 source head 9.5 keV), resulting in high density ion production. Subsequently, electrostatic voltages are applied to the drift tube to guide the reagent ions toward the center axis of the inlet, allowing them to interact with neutral molecules in the sample gas flow with a reaction time of approximately 300 ms. The sample gas is introduced into the center axis of the CI inlet through a ¾" stainless steel tube at a

flow rate of around 6 lpm (liter-per-minute). Ion-molecule reactions with $NO_3^-$ can occur through either proton abstraction and clustering (e.g. small dicarboxylic acids) or only by clustering (e.g. OVOCs, HOMs) (Field, 1968; Jokinen et al., 2012). Clustering reactions involve a sample molecule such as HOMs that generates a stable ion-molecule cluster $(NO_3)\cdot HX^-$ (Hyttinen et al., 2015). The product and reagent ions then enter the mass spectrometer through a critical orifice (diameter 0.3 mm), at a flow of approximately 0.8 L min$^{-1}$, and are subsequently focused through a series of ion optics as they move towards

the time-of-flight mass spectrometer (ToFMS) region where they are separated and detected according to their time of flight in the ToF chamber. The time of flight is then processed and converted to the mass-to-charge ratio of the ion in question. The instrument is characterized by a moderate mass resolution of about 3700 m/Δm. The ToFMS is operated at 16.7 kHz frequency with a 60 μs ToF extraction period. The mass spectrum range is 7−1126 Th. Data are collected at a 1-second time resolution and averaged over a 1-min intervals. The mass (m/z) calibration, a crucial step in data processing, is usually performed with

respect to calibrant peaks. Typically, the 3 reagent ion peaks, $NO_3^-$, $(HNO_3)\cdot NO_3^-$ and $(HNO_3)_2\cdot NO_3^-$ (m/z=61.99; 124.98; 187.98 respectively) are used. However, it is important to have several reference peaks well distributed along the covered m/z range. Fluorinated organic compounds appeared clearly as contaminants that likely originated from Teflon® sampling lines used in early experiments and therefore appeared in the mass spectra. This phenomenon is well-known in the use of $NO_3^-$ ToFCIMS (Ehn et al., 2012). To make use of this, several perfluorinated organic acids covering the upper m/z range were

added continuously to the instrument (see Table 2), so that the mass calibration could cover a wider mass range leading to an improvement of the mass calibration with a mass accuracy of less than 10 ppm.

**Table 2: List of perfluorinated organic compounds chosen for mass calibration covering a wide range of m/z.**

| Compound | MW (g mol$^{-1}$) | Chemical Formula | Form of detection (m/z of detection) |
|---|---|---|---|
| Perfluoropropionic acid | 164.03 | $C_2F_5COOH$ | $C_3HF_5O_2\cdot(NO_3)^-$ (225.978)> $C_3F_5O_2^-$ (162.982) |
| 2,3,4,5,6-pentafluorobenzoic acid | 212.07 | $C_6F_5COOH$ | $C_7HF_5O_2\cdot(NO_3)^-$ (274.058)>>> $C_7F_5O_2^-$ (211.07) |
| Perfluoroheptanoic acid | 364.06 | $C_6F_{13}COOH$ | $C_7HF_{13}O_2\cdot(NO_3)^-$ (426.048)> $C_7F_{13}O_2^-$ (363.06) |
| Perfluorononanoic acid | 464.08 | $C_8F_{17}COOH$ | $C_9HF_{17}O_2\cdot(NO_3)^-$ (526.068)> $C_9F_{17}O_2^-$ (463.08) |
| Perfluoroundecanoic acid | 564.09 | $C_{10}F_{21}COOH$ | $C_{11}HF_{21}O_2\cdot(NO_3)^-$ (626,068) >>> $C_{11}F_{21}O_2^-$ (563.09) |

**> Slightly higher than        >> Significantly higher than**





## 2.2 Experimental approaches

### 2.2.1 Approach 1− Organic Vapor Pressure Quantification

Following Approach 1, organic compounds (OC) that are in the form of solid powders (liquid form for pyruvic acid), are placed in ¼ inch outside diameter stainless steel tubes. The compounds are confined by filters, composed of stainless steel grids or glass wool, on each end that serve to keep solid/liquid materials from entering the instrument (see Figure 1). This device is designated as the source tube (ST), and is heated using a temperature controlled heating taped to a fixed temperature whose value was chosen depending on the compound (ranging from 20 to 90°C). Heating serves to increase the vapor pressure

of the compound. The choice of temperature varies with the compound being studied and is determined through experimental testing. These tests involve gradually increasing the temperature while measuring with the instrument, and avoiding the decomposition of the compound. It was observed that the signals from the reagent ions began to diminish beyond a certain temperature, both in the presence and absence of a sample. This somewhat restricted the range of compounds that could be employed with this method.

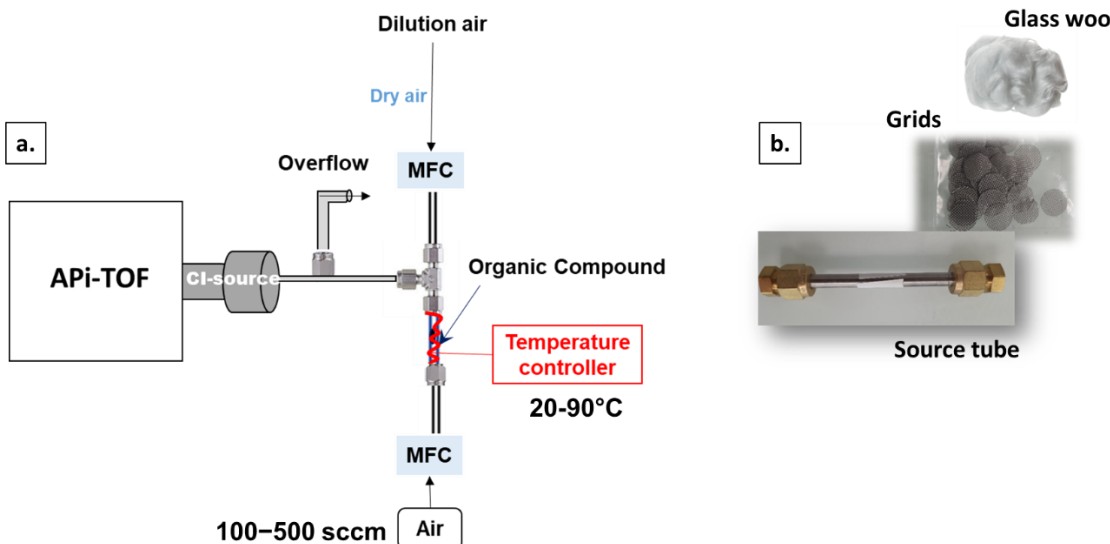


**Figure 1: a. Experimental set-up used for NO₃⁻ ToFCIMS calibration using organic vapor pressure quantification. b. Stainless steel grids, glass wool and tube containing the organic compound.**

The vaporized compounds are transferred out of the ST by a regulated flow of synthetic air (100 to 500 sccm) controlled by a calibrated mass flow controller (MFC). The flow is diluted by additional dry zero air that controls the final

concentration and provides excess flow to the inlet (see Figure 1.a).

Table 3 provides a summary of the tested organic compound calibrants for the NO₃⁻ ToFCIMS, consisting mainly of dicarboxylic acids. Additional compounds have been tested but didn't work. The instrument exhibited a measurable response to the following organic compounds: propanoic (PrA, purity 99.5%), pyruvic (PyA), lactic (LA), oxalic (OxA, purity ≥99.0%),





succinic (SucA, purity ≥99.5%), tartaric (TA, purity ≥99.0%), and malonic (MA, purity ≥99.0%) acids, along with 4-
nitrocatechol (4-NC, purity ≥96%) (the only nitrophenol tested).

**Table 3: Candidate organic compounds used to evaluate the sensitivity of the instrument. *Upon heating, this compound exhibits color changes, indicating decomposition.**

| Compound | MW (g mol$^{-1}$) | Chemical structure |
|---|---|---|
| Propanoic Acid | 74.08 | $C_3H_6O_2$ |
| Pyruvic Acid | 88.06 | $C_3H_4O_3$ |
| Oxalic Acid | 90.03 | $C_2H_2O_4$ |
| Lactic acid | 90.08 | $C_3H_6O_3$ |
| Malonic Acid | 104.06 | $C_3H_4O_4$ |
| Succinic Acid | 118.09 | $C_4H_6O_4$ |
| Tartaric Acid | 150.08 | $C_4H_6O_6$ |
| 4-nitrocatechol | 155.11 | $C_6H_5NO_4$ |
| Benzenesulfonic Acid$^*$ | 158.17 | $C_6H_6O_3S$ |

The concentrations of these compounds that were produced and injected into the ToFCIMS were calculated using Eq. (2).

$$[OC]_{(ppbv)} = \frac{F_{OC}(sscm) \times P_{vap}(Pa)}{(F_{OC}(sscm) + F_{diluent}(sscm)) \times P_{atm}(Pa)} \times 10^9 \,, \tag{2}$$

Where, [OC] is the concentration of the organic compound, in ppbv, $F_{OC}$ is the flow through the ST, in standard cm$^3$ min$^{-1}$
(sccm), $P_{vap}$ is the vapor pressure of the organic compound, in Pa, at the ST temperature employed, $F_{diluent}$ is the dilution flow,
in sccm, and $P_{atm}$ is the ambient pressure, in Pa.

The overall formula is multiplied by $10^9$ to convert the mixing ratio to ppbv. For these experiments, vapor pressures are
obtained either experimentally (see 2.2.2) or extracted from the literature.

The $P_{vap}$ of a compound at a specific temperature $T_2$ was calculated using the Clausius–Clapeyron Equation (Eq. (3))
by using a known value of the $P_{vap}$ at a reference temperature, $T_1$, and either the enthalpy of vaporization for a liquid or the
enthalpy of sublimation for a solid compound, as appropriate.

$$\ln\left(\frac{P_2}{P_1}\right) = \frac{\Delta H_{vap/sub}}{R} \times \left(\frac{1}{T_1} - \frac{1}{T_2}\right) \,, \tag{3}$$

Where, $P_1$ and $P_2$ are the vapor pressures at temperatures $T_1$ and $T_2$, respectively, and $\Delta H_{vap/sub}$ is the enthalpy of vaporization
or sublimation; R is the gas constant (8.314 J mol$^{-1}$ K$^{-1}$).

### 2.2.2 Experimental Determination of Vapor Pressure of Malonic Acid

A series of laboratory experiments was conducted to experimentally determine the vapor pressure (Pvap) of MA, which
is a solid at room temperature. The experimental procedure was as follows:





1- Air at a flow rate of 300 sccm was passed through the ST, which was heated to 50°C and connected to the ToFCIMS instrument.

2- The experiment was left to run for a duration of one week or longer, enabling the measurement of a detectable loss of mass using an analytical balance with an accuracy of 0.1 mg.

3- The vapor pressure of the compound was then deduced using Eq. (4), which is derived from the ideal gas law. This experimental procedure was repeated three times for accuracy and consistency.

$$\mathbf{P_{vap}} = \frac{\Delta\mathbf{m_{meas}(g) \times R \times T(K)}}{\mathbf{MW(g.mol^{-1}) \times V(L)}}, \tag{4}$$

Where, $P_{vap}$ is the vapor pressure, in atmospheres (atm), $\Delta m_{meas}$ is the weight loss (g), T is the temperature (K), V is the volume of air (L), found by multiplying the air flow rate over the source tube (lpm) by the time (min), and $R = 0.082057$ L atm mol$^{-1}$ K$^{-1}$.

### 2.2.3   Approach 2− Organic Sensitivity by FTIR Quantification

The experimental methodology of Approach 2 consists of the two main elements:

- The atmospheric simulation chamber (CSA) equipped with an in-situ FTIR (Fourier Transform Infrared) spectrometer instrument (Bruker Vertex 80).

- A vacuum line coupled to a bulb of known volume (0.3 L) which was used (see Figure 2) to prepare gaseous OCs from liquid standards to inject into the chamber.

Briefly, the CSA chamber (LISA, UPEC) is an atmospheric simulation chamber, which is a cylindrical Pyrex reactor (volume: 977 L, length: 6 m, diameter: 45 cm) designed for investigating atmospheric gas processes under controlled conditions. In addition, it is equipped with instrumentation for analysis using ultraviolet/visible and infrared spectroscopy (Doussin et al., 1997; Picquet-Varrault et al., 2005). The chamber is equipped with an efficient homogenization system, ensuring a mixing time of less than a 1 min. In our experimental studies, FTIR spectra were averaged for 5-minutes and covered the spectral range of 500-4000 cm$^{-1}$, with a spectral resolution of 0.5 cm$^{-1}$ and an optical path length of 214 m.

The basic principle of the vacuum line involves placing the compound in a glass finger connected to the vacuum line, which is plunged into a cold trap of liquid nitrogen. The vacuum is then applied to eliminate air and volatile impurities. The pump is then isolated and the finger brought back to room temperature allowing the evaporation of the compound into the bulb. The bulb pressure is measured and the concentration of the compound is calculated using Eq. (5).



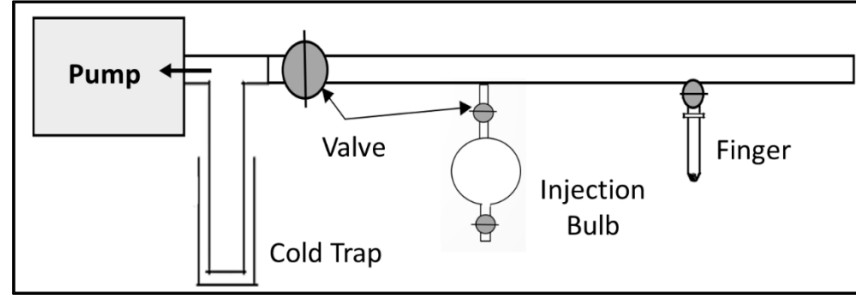

205

**Figure 2: Vacuum line for the preparation of the compounds injected into the simulation chamber.**

$$[OC]_{(ppbv)} = \frac{(P_{f,bulb} - P_{i,bulb}) \times V_{bulb}}{P_{CSA} \times V_{CSA}} \times 10^9, \tag{5}$$

Where, $P_{i,bulb}$ is the initial bulb pressure (limit vacuum around $10^{-4}$ mbar), $P_{f,bulb}$ is the final bulb pressure, $P_{CSA}$ is the CSA chamber pressure, $V_{bulb}$ and $V_{CSA}$ are the bulb and CSA chamber volumes, respectively.

The principle of this approach (see Figure 3) involves several steps, as follow:

1- The CSA is first filled with nitrogen gas to slightly exceed ambient pressure by about 5 mbar;

2- The liquid organic under study is introduced into the chamber by passing synthetic air through the bulb;

3- FTIR spectra are recorded and the stabilization of the corresponding organic signal is ensured;

4- The chamber contents are diluted using a pumping system and by connecting the $NO_3^-$ ToFCIMS to the chamber, using a heated ¼ inch diameter stainless steel line (maintained at ~40°C), yielding a total dilution flow of approximatively 23 lpm;

5- The pressure inside the chamber is maintained by continuously introducing $N_2$ into the chamber;

6- The normalized ion counts of the OC, obtained by the $NO_3^-$ ToFCIMS, and the chamber concentrations derived from the FTIR (using reference IR spectrum for the OC) are used to determine a calibration factor for the compound being studied.

This approach was only appropriate for our liquid compounds that are characterized by higher vapor pressures compared to the solid and can be introduced in sufficient quantities to compensate for rather high detection limits of the FTIR.


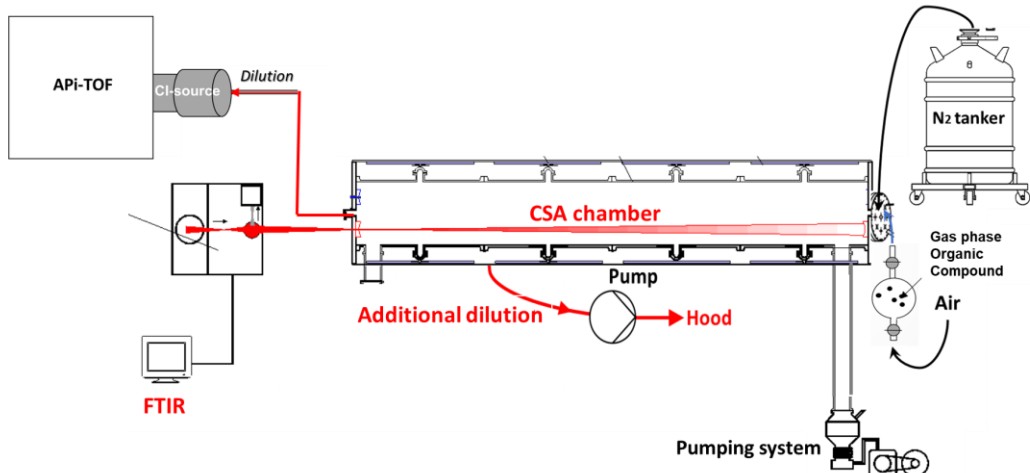

**Figure 3: The experimental setup to deduce the sensitivity of $NO_3^-$ ToFCIMS to an organic compound from its concentration derived by FTIR spectrometry using the CSA chamber.**

The FTIR spectra were processed using Analysis of Infrared Spectra (ANIR) software, which consists of a classic fitting routine of the spectra. For that, reference spectrum of the compound in question must be available with known optical path L(cm), reference concentration of the absorbing species C (molecule.cm$^{-3}$), and thus the effective absorption cross sections as a function of wavelength or wavenumber, $\varepsilon(\lambda)$.

The total uncertainty (in %) mentioned in the subsequent results encompasses all the uncertainties associated with the components used to calculate $C_X$ (e.g. ion signals, MFC, $P_{vap}$, $\Delta H_{vap/sub}$) (Supplement S.1).

### 2.2.4    Experimental Calibration with Sulfuric Acid

To make accurate comparisons between our laboratory studies, field results, and reports in the literature, we also calibrated the $NO_3^-$ ToFCIMS instrument using $H_2SO_4$ as calibrant, which is a procedure employed in several studies (e.g. Rissanen et al., 2014; Mutzel et al., 2015; Pullinen et al., 2020). This calibration procedure includes generating a specific amount of OH· in presence of excess $SO_2$ that reacts to form $H_2SO_4$, following reactions R. (1), R. (2), R. (3) and R. (4). A calibration unit was used that was developed based on the work of Kürten et al. (2012). It consists of a mercury lamp providing 184.9 nm UV radiation and a quartz glass tube to which is added a flow of humidified air. OH· radicals are generated from the photolysis of water vapor by the ultraviolet radiation, which is followed by these reactions:

$$H_2O + h\nu\,(184.9\ nm) \rightarrow OH + H \qquad\qquad \textbf{R. (1)}$$

$$SO_2 + OH + (M) \rightarrow HSO_3 \qquad\qquad \textbf{R. (2)}$$

$$HSO_3 + O_2 \rightarrow SO_3 + HO_2 \qquad\qquad \textbf{R. (3)}$$

$$SO_3 + 2H_2O \rightarrow H_2SO_4 + H_2O \qquad\qquad \textbf{R. (4)}$$





Three calibration setups were constructed for the $NO_3^-$ ToFCIMS instrument (Table 4). In the first setup, the
calibration source was connected to the instrument using a Swagelok tee (to overfill the inlet). The second setup involved
connecting the calibration unit to the ToFCIMS through a 1-meter ¾-inch stainless steel tube (also with a tee to overfill). This
line was used to sample ambient air during field campaigns. Finally, the third setup replicated the apparatus employed in the
calibration approach described in section 2.2.1 and Figure 1, which was used to apply the calibration Approach 1. In this
configuration, the sulfuric acid calibration source replaced the ST, and notably, no heating was applied. The first two setups
were designed to assess the wall loss of sulfuric acid in a 1-meter sampling tube. The third setup was conducted to collect data
to compare the calibration factors obtained from the $H_2SO_4$ source and the organic compounds that were tested.

**Table 4: Experimental Setups for H₂SO₄ Calibration Source.**

| Setup N° | Inlet sampling flow (slpm) | Comments |
|---|---|---|
| 1 | 8 | connected to the inlet using a Swagelok tee |
| 2 | 8 | connected to a 1 m length tube (ACROSS campaign setup) |
| 3 | 6 | connected to apparatus used in calibration Approach 1 (heated ST) |

To conduct a calibration experiment, a range of $H_2SO_4$ concentrations were generated. The $SO_2$ concentration was
kept constant while varying the $H_2O$ concentrations which results in different OH concentrations. $SO_2$ was delivered from a
ALPHAGAZ™ Mix cylinder (9.04 ppm in a mix of $N_2$ and $O_2$) to create a mixing ratio of about 770 ppbv in the source. To
prevent absorption of UV light by ambient $O_2$ and $H_2O$ vapor in the space between the mercury lamp and the quartz tube, the
unit was purged with dry $N_2$ (ALPHAGAZ 2). The $H_2O$ vapor mixture was generated by passing an air flow through an
ultrapure water bubbler.

The $H_2O$ vapor mixing ratio and OH concentration are determined using Eq. (6) and (7), respectively.

$$[H_2O] = \frac{Q_{H_2O}}{Q_{H_2O} + Q_{SO_2} + Q_{air} + Q_{N_2}} \times \frac{p_{sat}(T) \times N_A}{R \times T} \tag{6}$$

Where, $Q_{H2O}$, $Q_{SO2}$, $Q_{air}$ and $Q_{N2}$ are the flow rates of humidified air, $SO_2$ mixture, dry air and $N_2$, respectively, $p_{sat}$ (T) is the
saturation vapor pressure of water, at temperature T, calculated using the Antoine Equation (Bridgeman & Aldrich, 1964), $N_A$
is Avogadro's Number, and R is the ideal gas constant .

$$[OH] = I \times t_r \times \sigma_{H2O} \times \Phi_{H2O} \times [H_2O] \tag{7}$$

Where, I is the photon flux (photons $cm^{-2}$ $s^{-1}$), $t_r$ is the illumination time (s). The quantity $I \times t_r$ is determined from actinometry
experiments based on the photolysis of $N_2O$ producing $NO_x$ (Kürten et al., 2012), $\sigma_{H2O}$ is the absorption cross section of water
vapor at 184.9 nm (Cantrell et al., 1997), $\Phi_{H2O}$ is the photolysis quantum yield assumed equal to 1, and $[H_2O]$ is the
concentration of water calculated from Eq. (6).

The various parameters and the values used in this study are listed in Table 5.





The concentrations of $H_2SO_4$ were estimated by assuming that all OH radicals produced react with $SO_2$. The $H_2SO_4$ calibration factors, denoted C(sulfuric) were calculated using Eq. (1).

**Table 5: Parameters employed in the $H_2SO_4$ source used during calibration experiments of the $NO_3^-$ ToFCIMS.**

| Parameter | Value | Units |
|-----------|-------|-------|
| $Q_{H2O}$ | 10-300 | slcm |
| $Q_{N2}$ | 0.098 | slpm |
| $Q_{SO2}$ | 1.08 | slpm |
| $Q_{air}$ | 11.4 | slpm |
| $p_{sat}$ (T) | 0.02771 | atm |
| $N_A$ | $6.022 \times 10^{23}$ | molec $cm^{-3}$ $mol^{-1}$ |
| T | 23 | °C |
| R | 0.08206 | L atm $mol^{-1}$ $K^{-1}$ |
| $I \times t_r$ | $2.1 \times 10^{11}$ | photons $cm^{-2}$ |
| $\sigma_{H2O}$ | $7.22 \times 10^{-20}$ | $cm^2$ molecule |
| $\Phi_{H2O}$ | 1 | - |

## 3    Results

### 3.1  Outcomes from Approach 1 (Heated ST)

**3.1.1 Pyruvic acid (PyA)**

Two experiments were conducted by putting a piece of glass wool with small amount of pyruvic acid (monocarboxylic acid, $C_3H_4O_3$) in the ST. The Pvap(PyA) at a specific T was calculated using Eq. (3) with a literature value for the standard molar enthalpy of vaporization ΔH vap(PyA) at 298.15 K of $53.6 \pm 2.1$ kJ $mol^{-1}$ (Emel'yanenko et al., 2018). The calculation used a Pvap of 289.9±7.3 Pa at a temperature of 308.2 K (Emel'yanenko et al., 2018). The normalized ToFCIMS signals of $C_3H_4O_3$
showed a linear increase with the flow through to the ST ($R^2$=0.98; see Figure 4). The C(PyA) values from both experiments yielded an average of $4.64 \times 10^{15}$ molecule $cm^{-3}$ with 5% of total uncertainty. By comparing this result with the values reported in the literature (Table 1), one can conclude that $NO_3^-$ ToFCIMS exhibit low sensitivity to $C_3H_4O_3$ despite its high O/C ratio. The ratio between ions of the deprotonated form ($C_3H_3O_3^-$; m/z 87.0087) and the ion of the cluster forms ($C_3H_4O_3 \cdot NO_3^-$; m/z 150.0044) is about 0.56.

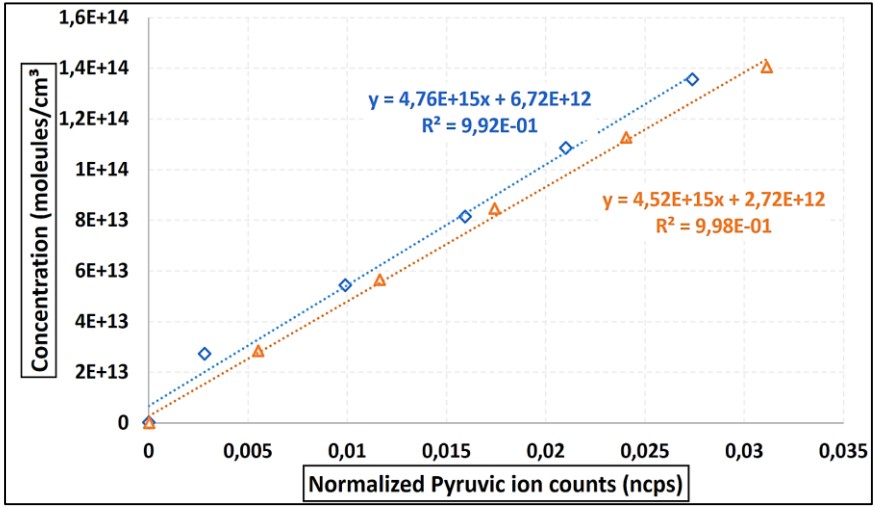


**Figure 4: NO₃⁻ ToFCIMS sensitivity to pyruvic acid derived from the linear fit to the injected concentration versus the pyruvic ion signals normalized to the total ion count of the reagent ions (ion ratio) for the two conducted experiments.**

### 3.1.2    Oxalic acid (OxA)

Several experiments were performed to evaluate the sensitivity of the instrument towards oxalic acid ($C_2H_2O_4$) using Approach
1 (Heated ST). An average of the solid Pvap(OxA) (298 K) values reported in the literature was used in Eq. (3) (Pvap$_{avg}$= $1.89\pm0.8 \times10^{-2}$ Pa) (Noyes & Wobbe, 1926; Bradley & Cotson, 1953; Wit et al., 1983; Booth et al., 2010). The Pvap at the experimental T was calculated according to Eq. (3) by taking the average of the published sublimation enthalpies $\Delta$Hsub(OxA)= $91\pm9$ kJ mol$^{-1}$ (average taken from Bilde et al., 2015). Figure 5 shows the C(OxA) obtained. The average value obtained for C(OxA) was $1.16\times10^{13}$ molecule cm$^{-3}$ with 44% of total uncertainty. This value is about 3 orders of magnitude
greater than the calibrations values reported in the literature for HOMs (meaning less sensitive). Yet, it is more than 2 orders of magnitude less than the value reported for malonic acid (Table 1). Once again, despite its high O/C ratio, the results suggest that the NO₃⁻ ToFCIMS exhibits lower sensitivity towards $C_2H_2O_4$ but demonstrates better sensitivity than $C_3H_4O_3$. The ratio between ions of the deprotonated form ($C_2HO_4^-$; m/z 88.9880) and the ion of the cluster forms ($C_2H_2O_4\cdot NO_3^-$; m/z 151.9836) is about 0.14.



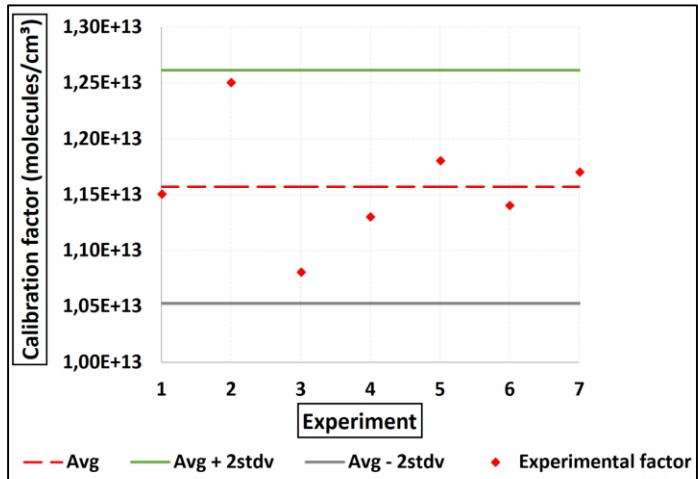


**Figure 5: Oxalic calibration coefficients obtained showing the mean (red line) and 95% confidence intervals (green and gray lines). The red symbols depict the calibration factors obtained from each experiment.**

### 3.1.3    Succinic acid (SucA)

Several experiments were conducted to evaluate the response of the instrument to succinic acid ($C_4H_6O_4$). The solid

Pvap(SucA) (298 K) is equal to $(7.7\pm5.0)\times10^{-5}$ Pa from the review of Bilde et al. (2015). Pvap(T) has also been calculated according to Eq. (3) by taking the average of the published sublimation enthalpies $\Delta$Hsub(SucA) = $(115\pm15)$ kJ mol$^{-1}$. An average value of C(SucA) = $1.65\times10^{13}$ molecule cm$^{-3}$ was achieved with about 66% of total uncertainty. Figure 6 shows the C(SucA) obtained from four successful tests. They are close to the one obtained for $C_2H_2O_4$. It is still approximately 3 orders of magnitude greater (meaning less sensitive) than the values reported for $H_2SO_4$ and the organic calibrants in the literature

(Table 1). This indicates that the $NO_3^-$ ToFCIMS exhibits a rather low sensitivity towards $C_4H_6O_4$ in comparison to $H_2SO_4$ detection. The ratio between ions of the deprotonated form ($C_4H_5O_4^-$; m/z 117.0193) and the ion of the cluster forms ($C_4H_6O_4 \cdot NO_3^-$; m/z 180.0149) is about 0.16.





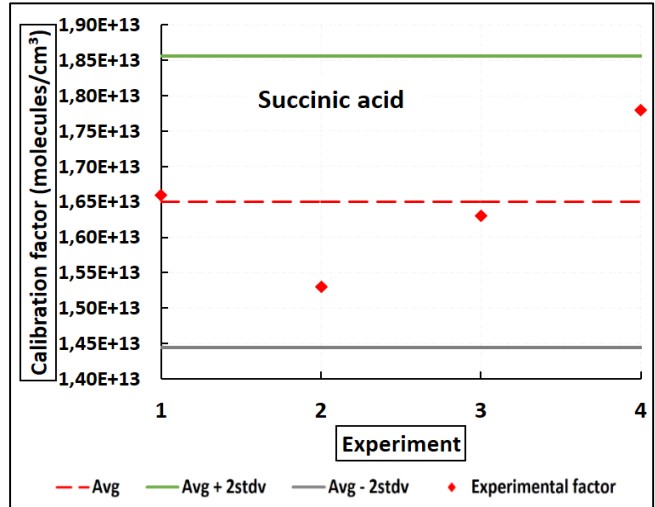

**Figure 6: Succinic acid calibration coefficients obtained. The red symbols depict the calibration factors obtained from each experiment performed.**

### 3.1.4 Malonic acid (MA)

Following the procedure described in section 2.2.2, an experimental mean value of Pvap(MA) (323 K) = (1.48±0.15) ×10$^{-2}$ Pa was obtained. Pvap(MA) (298 K) = 4.50×10$^{-4}$ Pa was determined using Eq. (3) employing the average of three published sublimation enthalpies ΔHsub(MA) = (111.8±14) kJ mol$^{-1}$ (Ribeiro da Silva et al., 1999; Booth et al., 2010; Cappa et al., 2008). This experimental value for the vapor pressure of malonic acid is comparable to the average of Pvap(MA) (298 K) =4.88×10$^{-4}$ Pa obtained in these studies employing eq. 4 and method described in section 2.2.2, with a relative difference of 7.7%. Our experimental value was used to estimate the calibration factor for $C_3H_4O_4$.

An average value of C(MA)= 4.27×10$^{12}$ molecule cm$^{-3}$ was achieved with about 30% of total uncertainty (Figure 7). This value is about two orders of magnitude greater than that the one reported by Krechmer et al. (2015) and Massoli et al. (2018) (Table 1) but lower than the calibration factor values obtained for $C_2H_2O_4$, $C_3H_4O_3$ and $C_4H_6O_4$. This indicates that the $NO_3^-$ ToFCIMS exhibits higher sensitivity towards $C_3H_4O_4$ compared to the other compounds that were tested. Nevertheless, there is a lack of adequate evidence to elucidate these discrepancies. The ratio between ions of the deprotonated form ($C_3H_3O_4^-$; m/z 103.0036) and the ion of the cluster forms ($C_3H_4O_4·NO_3^-$; m/z 165.9993) is approximatively 0.17.





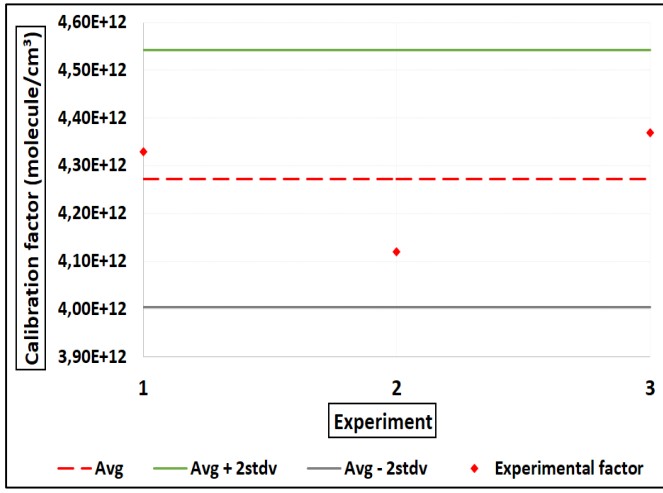

**Figure 7: Malonic calibration coefficients obtained within a 95% confidence interval. The red symbols depict the factors obtained from each experiment performed.**

### 3.1.5 Tartaric acid (TA)

For calibration of the instrument to tartaric acid ($C_4H_6O_6$), the Pvap(TA) (298 K)=(1.79±0.72)×10$^{-4}$ Pa was taken from Booth et al. (2010) who reported the only experimentally obtained values of Pvap(TA) and ΔHsub(TA). Similarly with the other

molecules studied, Pvap(TA) (T) was calculated using Eq. (3) with ΔHsub(TA) = (68±10) kJ mol$^{-1}$ (Booth et al., 2010). The average value obtained in this study for C(TA) is $5.84×10^{12}$ molecule cm$^{-3}$ with an estimated 43% total uncertainty. Figure 8 shows the C(TA) values obtained. This value is similar to that obtained for $C_3H_4O_4$ in this study. However, it is still approximately two orders of magnitude higher than the values reported in the literature for $H_2SO_4$ which is used for HOMs. The ratio between ions of the deprotonated form ($C_4H_5O_5^-$; m/z 149.0091) and the ion of the cluster forms ($C_4H_6O_6 \cdot NO_3^-$; m/z

212.0048) is approximatively 1.

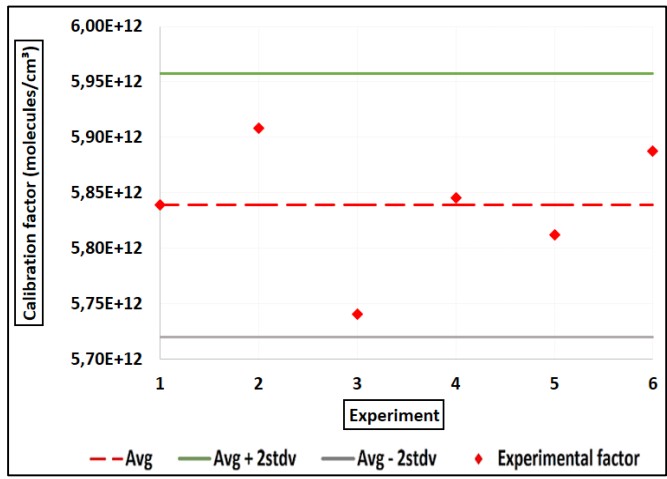

**Figure 8: Tartaric calibration coefficients obtained within a 95% confidence interval. The red symbols depict the factors obtained from each experiment performed.**



### 3.1.6    4-Nitrocatechol (4-NC)

The Pvap(4-NC) (313 K) equals $(1.49\pm0.0.55) \times10^{-3}$ Pa which was determined experimentally in the laboratory following the approach used for Pvap(4-NC) (323 K), described in section 2.2.2. Using ΔHsub(4-NC) = $(121.1\pm1.4)$ kJ mol$^{-1}$ (da Silva et al., 1986), Pvap(4-NC) (298 K) can be determined using Eq. (3). An average value of C(4-NC) = $1.49\times10^{11}$ molecule cm$^{-3}$ was obtained with estimated 16% total uncertainty (Figure 9). Among all the tested organic compounds, 4-nitrocatechol demonstrates the lowest $C_X$, indicating that, of the compounds studied, the instrument is more sensitive towards this molecule.

However, even with this better sensitivity, $C_6H_5NO_4$ still exhibits values approximately one order of magnitude higher than those reported the literature for $H_2SO_4$.The ratio between ions of the deprotonated form ($C_6H_4NO_4^-$; m/z 154.0145) and the ion of the cluster forms ($C_6H_5NO_4 \cdot NO_3^-$; m/z 217.0102) is approximatively 0.7.

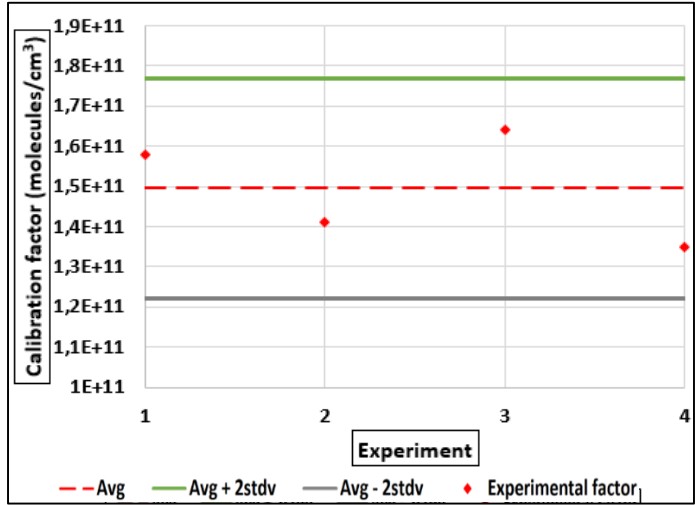

**Figure 9: 4-Nitro-catechol calibration coefficients obtained within a 95% confidence interval. The red symbols depict the factors**
**obtained from each experiment performed.**

### 3.2 Outcomes from Approach 2 (CSA)

Following approach 2, two experiments were carried out by adding pyruvic acid to the CSA chamber. Figure 10 displays the time series of PyA concentrations as determined by FTIR, and the corresponding normalized ion signals from the $NO_3^-$ ToFCIMS for the two experiments, labelled as Experiment 1 and Experiment 2, are shown. The yellow shaded regions in the
figure represent the periods during which dilution was introduced into the chamber.



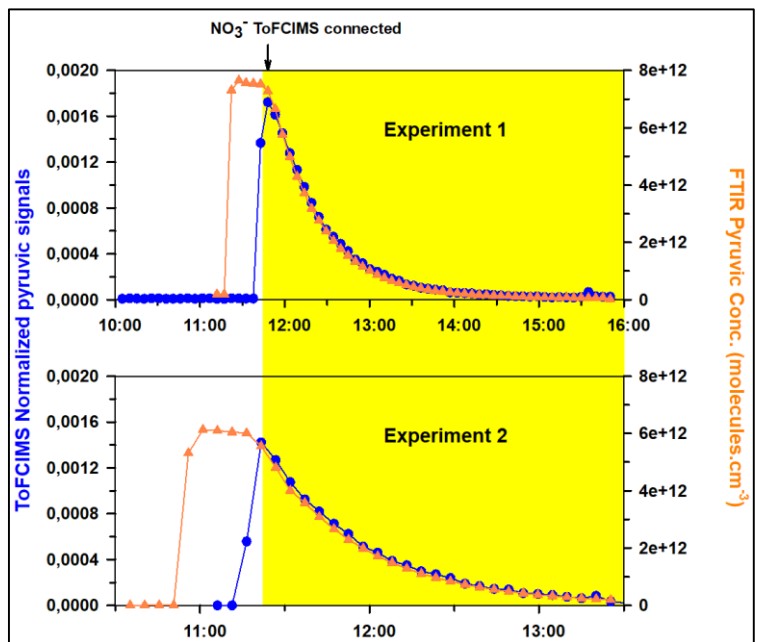

**Figure 10: Time series of pyruvic acid concentrations obtained by FTIR and the corresponding normalized ion signals from the NO$_3^-$ ToFCIMS.**

Furthermore, when fitting the pyruvic concentrations measured via FTIR against the normalized PyA ion signals acquired

from ToFCIMS, the resulting slope corresponds to the pyruvic acid calibration factor, denoted as C(PyA) (see Figure 11). Individual values can be calculated using Eq. (8).

$$C(PyA) = \frac{[PyA]_{FTIR} \, (molecules \, cm^{-3})}{Ion \, ratio_{PyA}} \qquad (8)$$

Where, C(PyA) is the calibration factor, in molecules cm$^{-3}$/ion ratio, [PyA]$_{FTIR}$ are the concentrations obtained from the FTIR

analysis, in molecules cm$^{-3}$, and Ion ratio$_{PyA}$ are the normalized ion signals for pyruvic acid obtained with the ToFCIMS.

The average value of C(PyA) from these experiments is $(3.81 \pm 0.03) \times 10^{15}$ molecule cm$^{-3}$/ion ratio. The relative difference between this value and the one obtained by Approach 1 is 18%. This difference could be explained by various factors including uncertainties on the values of Pvap(PyA) and ΔHsub(PyA) used in the calculation of Approach 1 and the uncertainties in the IR reference spectrum employed in Approach 2. However, in both cases, the C(PyA) value obtained in this study is about 5 orders of magnitude greater than the factors found in the literature using H$_2$SO$_4$ as calibrant (Table 1) and confirming that our

NO$_3^-$ ToFCIMS exhibits low sensitivity towards C$_3$H$_4$O$_3$.





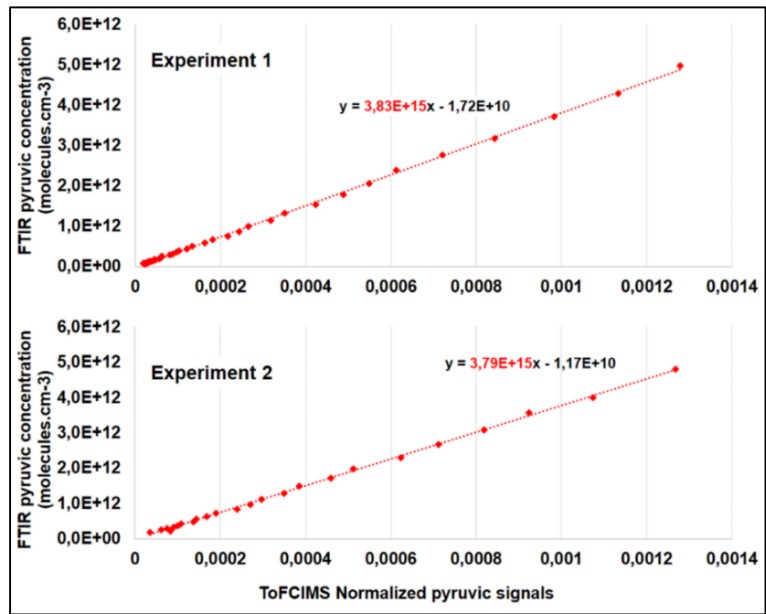

**Figure 11: FTIR pyruvic acid concentration vs normalized pyruvic acid signals of ToFCIMS. The red dashed lines are the fitted trend lines. The slopes equal the calibration factor for each experiment.**

Table 6 below provides a summary of the calibration factors obtained for the small dicarboxylic acids and 4-nitrocatechol. The

findings highlight a significant variability in the calibration factors, illustrating that the sensitivity of the $NO_3^-$ ToFCIMS is compound-specific, particularly for these small OVOCs.

**Table 6: Summary of the calibration factors resulted for the organic compounds acid measured with the $NO_3^-$ ToFCIMS. *Following Approach 1 (Heated ST); **Following Approach 2 (CSA)**

| Compound | $C_x$ (molecules cm$^{-3}$) |
|---|---|
| Pyruvic Acid | $4.64 \times 10^{15}$* <br> $3.81 \times 10^{15}$** |
| Succinic Acid | $1.65 \times 10^{13}$* |
| Oxalic Acid | $1.16 \times 10^{13}$* |
| Tartaric Acid | $5.84 \times 10^{12}$* |
| Malonic Acid | $4.27 \times 10^{12}$* |
| 4-nitrocatechol | $1.49 \times 10^{11}$* |

**3.3 Outcomes from calibration with sulfuric acid**

Table 7 summarizes the sulfuric acid calibration factors C(sulfuric) obtained using the different experimental setup. The C(sulfuric) values obtained from the three setups are within the range reported in the literature (0.165-6) $\times 10^{10}$ molecules cm$^{-3}$/ion ratio (see Table 1). Comparing setups 1 and 2 revealed a loss of approximately 33% in $H_2SO_4$ levels along the 1-meter sampling line.





Furthermore, we notice that the C(sulfuric) obtained from setup 3 differs significantly from those obtained for organic acids
but reveal a loss of 65% with the setup used. This loss cannot explain alone the differences observed with the literature for the
different OVOC tested and strengthen the hypothesis of sensitivity that are compound specific for this instrument.

**Table 7: The calibration factor deriving from three experimental setups.**

| Setup N° | C(sulfuric) (molecules cm$^{-3}$) |
|---|---|
| 1 | $2.82\times10^9$ |
| 2 | $4.22\times10^9$ |
| 3 | $8.07\times10^9$ |

## 4    Conclusion

Instrument calibration is a crucial step in ensuring the accuracy and reliability of analytical tools. Typically, the $NO_3^-$
ToFCIMS instrument is calibrated using sulfuric acid, and the resulting calibration factor C(sulfuric) is used to quantify all
detected species, including HOMs. In our efforts to find more suitable and reliable organic calibrants, we implemented
calibration procedures for the $NO_3^-$ ToFCIMS instrument to assess its sensitivity and linearity in detecting various
commercially available organic compounds.

The tested organic compound calibrants for the $NO_3^-$ ToFCIMS, are summarized in Table 3. Note that additional
compounds have been tested and were not detected by the instrument. This could possibly be due to either the instrument's
lack of sensitivity towards them or the need to develop more sophisticated methods to generate gas-phase standard mixtures
of low-volatility compounds. It is also possible that higher heating temperatures could be required to generate them in the
gaseous phase, but we found that the maximum temperatures that could be used without changing the instrument's performance
limited further increasing the source tube temperatures.

Our studies demonstrate substantial variability in the calibration factors (Table 6) obtained for the small dicarboxylic
acids and 4-nitrocatechol. Notably, 4-nitrocatechol exhibited the highest sensitivity, followed by malonic acid, tartaric acid,
oxalic acid, succinic acid, with pyruvic acid being the least sensitive. This shows that the sensitivity of the $NO_3^-$ ToFCIMS is
dependent on the specific structure of organic compound. Therefore, relying on a single calibration factor obtained from $H_2SO_4$
does not seem to be appropriate for quantifying all species detected using this technique. The calibration factor for pyruvic
acid showed good agreement between Approach 1 and 2 with a relative difference of 18%. We observed that the calibration
factor for malonic acid is approximately two orders of magnitude higher than values reported in the literature without any
apparent explanation. When considering all the $C_X$ values in Table 1, an average value of $2.02\times10^{10}$ molecules cm$^{-3}$ is obtained
($\sigma = 1.96\times10^{10}$ molecules cm$^{-3}$). This average is roughly one order of magnitude less than the one obtained for 4-nitrocatechol
by our instrument, more than 2 orders of magnitude less than malonic and tartaric acid, 3 orders of magnitude lower than oxalic
and succinic acid and more than 5 orders of magnitude less than pyruvic acid. The tested compounds are probably not suitable
to account for HOMs calibration factors because of their oxidation state or chemical structure which differ from the one of



HOMs. Furthermore, the relative contribution of various ionization reaction pathway cannot explain the differences observed for $C_X$ between the various OVOCs tested.

Additionally, the conventional calibration method for the $NO_3^-$ ToFCIMS using $H_2SO_4$ was applied following an approach similar to that in Kürten et al. (2012). This calibration was implemented using three different setups in the laboratory (Table 4), with the calibration factors obtained ($2.83$-$8.08 \times 10^9$ molecules cm$^{-3}$) are within the reported range in the literature ($0.2$-$6 \times 10^{10}$ molecules cm$^{-3}$) excluding an instrumental malfunctioning as plausible explanation for disagreement observed between $C_X$ for OVOCs determined in this study and $C_X$ from literature. A comparison between setups 1 and 2 indicated a loss of approximately 33% in $H_2SO_4$ levels along the 1-meter sampling line. Therefore, difference in the loss at the surfaces for HOMs may also lead to differences in the calibration factors.

Comparatively, the C(sulfuric) values derived from setup 3 differ substantially from those obtained for organic acids using our calibration approaches 1 and 2. The C(sulfuric) value is employed to quantify HOMs in laboratory and field measurements. This choice also allows for comparisons with other reports in the literature that quantified HOMs using a calibration factor derived from sulfuric acid. It should be noted that HOMs concentrations calculated this way should be considered as upper limits.

In summary, the calibration experiments have underscored the limitations of using sulfuric acid, for establishing calibration factors for quantificational detected compounds, especially small dicarboxylic acids. Without existing alternative, sulfuric acid is used to quantify all the species detected by the $NO_3^-$ ToFCIMS, including HOMs, by assuming similar ionization kinetic rate constants and comparable transmission efficiency. To ensure the relevance of such approach, it is crucial to identify and investigate organic compounds that more accurately represent the properties of HOMs, providing a more reliable and precise means of quantifying HOMs. The calibration factors obtained using these new compounds should be compared with those obtained using the sulfuric acid. Given that such compounds may not be readily available commercially, their synthesis in the laboratory becomes a necessity although difficult step needs to be undertaken.

It should be recognized, however, that each instrument has a unique set of operational parameters that dictate its performance, sensitivity, and detection capabilities. Factors such as the design and length of the sampling line, its diameter, and the sampling environment (e.g., temperature and humidity) can significantly impact the accuracy and representativeness of the analyzed sample. These factors may vary depending on the instrument's location, further highlighting the need for careful consideration.

*Author contributions.* SH, VM and CC initiated the research idea, with VM and CC providing oversight throughout the work's development. SH, VM, SH, BVP, MC and CC actively participated in the CSA chamber experiments. MR and AK facilitated the experimentations by providing the sulphuric acid calibration unit. SH conducted laboratory experiments, analyzed the results, and drafted the initial manuscript. CC and VM reviewed and refined the manuscript for clarity and coherence. All co-authors contributed substantially to the discussions and provided valuable feedback on the manuscript.

*Competing interests.* The contact author has declared that none of the authors has any competing interests.





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
