# Peer review of "A Nitrate Ion Chemical Ionization Atmospheric Pressure interface Time-of-Flight Mass Spectrometer (NO3- ToFCIMS): sensitivity study"

_Atmospheric Measurement Techniques, 2024_

## Author Response (AR1)

First of all, we would like to thank the reviewer for his comments on the manuscript. We did our best to address all the comments and modified the manuscript accordingly. We believe it has significantly improved the paper. The changes made to the revised manuscript are summarized below.

**Response to referee #1:**

*Summary and general comment:*

*Alage et al. present very interesting and important results obtained from laboratory experiments about the calibration of a fundamental instrument that have been extensively used in several studies. It is well known that VOC play a key role in atmospheric chemistry, forming highly oxygenated organic molecules and secondary organic aerosols. While the measurement of VOC is quite standardized the measurement of the HOM and especially their quantification is still very complicated and so far, there is not a proper method yet. As the authors say, at the moment, the HOMs estimation is made by a nitrate CIMS using H2SO4 as a reference. This is not ideal because these HOM are very different then sulphuric acid. Thankfully, in this study, the authors try to understand these issues by using commercially available small organic compounds, and they compare their signal with the one obtained for H2SO4.*

*With this study, the authors were able to prove that the sensitivity of the nitrate CIMS varies depending on the organic compound, showing once more that that a single calibration factor from sulfuric acid is not enough for the HOMs quantification.*

*Because of all these findings, and the needs for a calibration system for HOMs I think that this article is suitable for publication in AMT. Below I have added few minor comments.*

We sincerely thank Reviewer #1 for their insightful and encouraging assessment of our study. Their recognition of the significance of our investigation into evaluating the sensitivity of the NO3- ToFCIMS towards different commercially available small organic compounds is highly valued. We greatly value the reviewer's positive evaluation of our work and their recommendation for publication in AMT. We will diligently address the minor comments provided to further refine our manuscript.

1) *Introduction: This part is very well done with a detail explanation of the problem that the authors are investigating. However, I might suggest moving the second part (equation, table and text connected to it) in the method. So that the introduction goes directly to the point of the problem.*

We acknowledge the referee's feedback on the length of the introduction and agree on the need to streamline it to better focus on the key aspects of our study. To address this concern, we have

restructured the "Materials and Experimental Methods" section by subdividing the "2.1. The NO$_3^-$ ToFCIMS" part into two subsections:

- 2.1.1 Principle: In this subsection, we have retained lines 111-136 and have designated the corresponding Table 2 as Table 1.

- 2.1.2 Conventional Calibration Methods: We have relocated lines 70-79 from the introduction, as well as lines 88-102 along with Equation (1) and the former Table 1, to this subsection. Consequently, the table originally referenced as Table 1 is now denoted as Table 2.

We believe that these changes effectively address the referee's concerns while maintaining clarity and coherence throughout our manuscript.

2) *Approach 2: As far as I understood the nitrate CIMS is compared to the FTIR. Since the FTIR has high LOD the authors need to have high concentration in the chamber as they mention after line 220. I'm wondering if this value can be translated also at low concentration. Saying in other words how linear is the nitrate CIMS calibration over different concentrations.*

Indeed, generating low concentrations in the order of ppt in the chamber, detectable by FTIR, poses a significant challenge due to its high LOD. While this limitation restricts our ability to directly compare the nitrate CIMS calibration at such low concentrations, it's worth noting that our results from pyruvic acid using both Approach 1 (line 281) and Approach 2 (line 370) show consistency within the same order of magnitude, although different concentrations of pyruvic acid was used in approach 1 (Figure 4) and 2 (Figure 10-11). This tends to demonstrate the linearity of nitrate CIMS response at least in the range of concentrations explored by the two approaches.

3) *Line 236: Maybe this sentence needs to be rephrased:" A calibration unit was used that was developed based on the work of Kürten et al. (2012)."*

Modification has been done in the revised manuscript. The revised sentence now reads: "A calibration unit, developed based on the work of Kürten et al. (2012), was used."

4) *Figure 4. A bracket is missing in the Y-axe label. I would show the R2 just as a "normal" value (e.g. 0.992) and not in the exponential form. However, it's up to the author if they want to change that. X-axe label Pyruvic "acid", add the word acid. Also, in the caption.*

Modification has been done in the revised manuscript.

5) *Figure 5 caption: Add the word "acid" after the word "oxalic". This has been missing in several part of the paper. I would Pyruvic and Oxalic not followed by the word "acid" don't mean much.*

Modification has been done in the revised manuscript.

6) *Conclusions: This part is great but I would shorten it and make it more concise so that the message goes streight to the point. Here again is up to the authors to take up on this comment or not.*

Thank you for your input. We've revised the conclusions section to focus on essential points and eliminate unnecessary details, aiming for a more direct and concise presentation of our key findings.

**Response to referee #2:**

We appreciate the valuable feedback provided by the reviewer on our manuscript. We have diligently addressed each comment and made corresponding modifications to the manuscript. We believe that these revisions have substantially enhanced the quality and clarity of the paper. The changes made to the revised manuscript is provided below.

*HOMs are important precursors for secondary particles in the atmosphere and subject extensive studies in the past decade. Among various detection techniques, nitrate CIMS appears to be the most widely used instrument owing to its capacity of detecting the least volatile group of compounds in HOMs. An accurate measurement of HOM concentration is required for quantifying the potential contribution of HOMs to SOA or NPF, which however, have not been achieved due to the lack of an effective calibration method. In views of this, this study shows advances in the calibration of nitrate CIMS, which makes it certainly important in this research field. Thus, I would recommend it acceptance in AMT after my comments are addressed.*

We sincerely thank Reviewer #2 for their insightful and encouraging assessment of our study. Their recognition of the significance of our investigation into evaluating the sensitivity of the $NO_3^-$ ToFCIMS towards different commercially available small organic compounds is highly valued. We greatly value the reviewer's positive evaluation of our work and their recommendation for publication in AMT. We will diligently address the minor comments provided to further refine our manuscript.

*General comments*

*This manuscript have stressed, in a few places, that this study highlights that HOM sensitivity could vary by orders of magnitude. This is of course true, but not new. Trostl et al., (2016) and Hyttinen et al., (2015) both clearly clarified this issue. Similarly, it is well known that the calibration coefficient of sulfuric acid cannot be directly apply to HOM quantification, which merely gives an lower limit of HOM concentration. Therefore, I think this paper contributes to briefly quantify the possible magnitude of such underestimation, rather than show this fact. I hope some relevant statements can be revised accordingly.*

Thank you for highlighting the prior work by Tröstl et al. (2016) and Hyttinen et al. (2015). We acknowledge the existing literature on the variability of HOM sensitivity. Our intention was to emphasize the practical implications of this variability rather than introduce a new concept. We have revised the sentence in line 426 and 427 to better clarify this point, as follows: "This loss cannot explain alone the differences observed with the literature for the different OVOC tested and strengthen the studies made by Hyttinen et al. (2015) and Tröstl et al. (2016) showing the compound specific sensitivity of this instrument.

*Specific comments:*

7) *I think the title is somewhat misleading or overstated. This study tested several commercially available calibrants, which however, are not enough to say it is a systematic or full calibration study of nitrate CIMS.*

Thank you for your feedback. Our intention was not to imply a systematic or full calibration study of nitrate CIMS.
In response to your comment, we have carefully reconsidered the title, removing the word "calibration", which might have been misleading in the original title. The revised title now reads: "A Nitrate Ion Chemical Ionization Atmospheric Pressure interface Time-of-Flight Mass Spectrometer ($NO_3^-$ ToFCIMS): sensitivity study."
We hope this adjustment addresses your concern and clarifies the focus of our research.

8) *In the calibration of any compound, it is essential to show the linear regression between CIMS signals and injected concentrations (just as in Fig.4). I wonder why it is not the case for Fig.5-9.*

We agree on the importance of demonstrating the linear regression between CIMS signals and injected concentrations for each compound tested.
Given the number of experiments conducted for each compound, we were mindful of not overwhelming the main body of the manuscript with an excessive number of figures. However, we recognize the significance of providing comprehensive information.
In response to your suggestion, we have decided to incorporate all the requested figures into the Supplement (Section S2).

9) *In the Approach 1 calibration, do the authors consider the wall loss of these organic compounds, which will directly affect the calibration coefficient. For the significantly higher sulfuric acid calibration coefficient in setup 3 (Table 7), additional loss could rise from particle nucleation of sulfuric acid (that consumes sulfuric acid).*

Wall losses are indirectly considered in the various setups, thus are incorporated in the corresponding Cx values. However, the results of approach 3 demonstrate that these losses, estimated to be of 65% as an upper limit with setup of approach 1, cannot account for the discrepancies observed among the Cx values for the different organic compounds tested.

10) *Line 430. It should be the lower limit rather than the upper limit.*

Thank you for your clarification. It's crucial to acknowledge the assumptions made regarding the ionization efficiency of HOMs with nitrate ions, assumed to be similar to that of H2SO4, which is close to the collision limit. As highlighted by Ehn et al. (2014), if HOM do not efficiently charge with nitrate ions at their collision limit or if they have a lower stability in the clusters formed, there is a risk of underestimating their concentration. Therefore, the assumption serves as a lower limit for the HOM concentration. I have made the necessary modifications in the revised manuscript to reflect this consideration.

*Minor comments*

*I found a few places where the expression of terms appear to be inconsistent with the literature or this manuscript itself:*

*e.g., Line 35 "… in the formation and growth of secondary organic aerosols (SOA)". SOA is usually used in a single form (secondary organic aerosol); "growth" is preserved for particle growth in size (a later stage of new particle formation).*

Modifications have been done in the revised manuscript.

*Line 40-45 ELVOC stands for "extremely low-volatility organic compounds" instead of "extremely low volatile organic compounds". LVOC and ULVOC are likewise.*

Modifications have been done in the revised manuscript.

*Line 60 and 65. NO3- ToFCIMS or nitrate ToFCIMS, please pick one and keep consistent.*

Modifications have been done in the revised manuscript.

*Line 81 "can be sued to" should be "can be used to"?*

Modifications have been done in the revised manuscript.

*Line 129 "a 1-min intervals", where "a" should be removed.*

Modifications have been done in the revised manuscript.